# Active household contact screening for tuberculosis and provision of isoniazid preventive therapy to under-five children in Afghanistan

Said Mirza Sayedi[1]*, Mohammad Khaled Seddiq[2], Mohammad K. Rashidi[1], Ghulam Qader[1], Naser Ikram[3], Muluken Melese[4], Pedro G. Suarez[4]

1 Challenge TB Project, Management Sciences for Health, Kabul, Afghanistan, 2 National TB Program, Ministry of Public Health, Kabul, Afghanistan, 3 Office of Health and Nutrition, United States Agency for International Development, Kabul, Afghanistan, 4 Management Sciences for Health, Arlington, VA, United States of America

* msayedi1980@gmail.com

**Data Availability Statement:** All relevant data are within the manuscript and its Supporting Information files.

## Abstract

### Objectives

This observational study analyzed the performance of the National TB Control Program (NTP) in Afghanistan in household contact screening from 2011 to 2018 and its use as an entry point for isoniazid preventive therapy (IPT), as well as the IPT completion rates for children under age five.

### Methods

From 2011 to 2018, the Afghanistan NTP released guidelines for passive and active contact screening of bacteriologically confirmed TB cases. Health workers were trained in contact screening. Presumptive TB cases gave sputum for AFB smear microscopy; other diagnostics were used if patients could not produce sputum. Children under five (excluding those with active TB) were treated for latent TB infection. We calculated the yield and the number needed to screen and number needed to test to find a case of TB, as well as the rates of IPT initiation and completion.

### Results

From 2011 to 2018, 142,797 bacteriologically confirmed TB cases were diagnosed in Afghanistan. The number of household members eligible for screening was estimated to be 856,782, of whom 586,292 (81%) were screened for TB and 117,643 (20.1%) were found to be presumptive TB cases. Among the cases screened, 10,896 TB cases (all forms) were diagnosed (1.85%, 95% CI 1.82–1.89), 54.4% in females. The number needed to screen to diagnose a single case of TB (all forms) was 53.8; the number needed to test was 10.7. Out of all children under five, 101,084 (85.9%) were initiated on IPT, and 69,273 (68.5%) completed treatment.

**Funding:** The United States Agency for International Development (USAID) funded this study through the Challenge TB project under cooperative agreement number AID-OAA-A-14-00029. The contents of the article are the responsibility of the authors alone and do not necessarily reflect the views of USAID or the US government. The publication fee is covered by the USAID funded Sustainable Technical and Analytic Resources (STAR) project, through Public Health Institute (PHI).

**Competing interests:** None declared.

## Conclusions

Program performance in contact screening in Afghanistan is high, at 81%, and the yield of TB is also high—close to 10 times higher than the national TB incidence rate. IPT initiation and completion rates are also high as compared to those of many other countries but need further improvement, especially for completion.

## Introduction

Tuberculosis (TB) is a major global public health problem and represents 1 of the top 10 causes of death globally. In 2018, an estimated 10 million people developed TB disease, which caused death in 1.5 million people [1]. In 2018, a total of 7 million new TB cases were notified by national health authorities and reported to the World Health Organization (WHO), but more than 3 million TB cases were missed [1]. Although Afghanistan is not on WHO's list of the 30 high-TB-burden countries, the incidence of TB is as high as in some of the high-burden countries, with a rate of 189 per 100,000 population, which has remained the same since 2000, although treatment coverage has greatly improved, from 19% in 2000 to 69% in 2018. In 2018, Afghanistan reported 48,420 TB cases out of the estimated 70,000 incident cases [1].

One strategy to increase case detection is contact screening of bacteriologically confirmed TB index cases. In 2011, the National TB Control Program (NTP) of Afghanistan initiated passive household contact screening, whereby bacteriologically confirmed index TB patients are counseled to bring their contacts for screening, and active household contact screening was also implemented in selected provinces [2]. In 2014, active household contact screening was expanded to Kabul and in 2015 further expanded to another five provinces. Contact screening was also used as an entry point to implement isoniazid preventive therapy (IPT) for children under five.

Although contact screening is a high-yield intervention recommended by WHO [3], its yield differs based on the country disease burden, diagnostic methods used, and health workers' capacity. The reported yield of contact screening ranges from 656 per 100,000 population [4] to 1,788 per 100,000 population in Viet Nam [5], 914 per 100,000 in Peru [6], 610 per 100,000 in Ethiopia [7], and 2,200 per 100,000 in Pakistan [8]. In an Afghanistan study of household contact screening that reported only on smear-positive cases, the yield was 1,880 per 100,000 population in the first year and decreased to 1,400 per 100,000 in the second year of intervention [9]. The yield was almost 10 times higher than the national estimated incidence of 189 per 100,000 population for Afghanistan [1].

This article analyzes the routine contact screening performance of the Afghanistan NTP from 2011 to 2018, with a focus on how contact screening helped expand IPT for children under five.

## Methods

### Setting

The Afghanistan NTP was initiated in 1954 by the Ministry of Public Health (MOPH), and the directly observed treatment short course (DOTS) strategy was introduced in 1997 but not rolled out nationwide until 2002 due to the security situation in the country [2]. The TB control program is integrated into primary and secondary health services through the Basic Package of Health Services and Essential Package of Hospital Services. Currently 2,627 health

facilities in the country provide DOTS services, and 883 laboratories carry out sputum microscopy (MOPH, National TB Report 2018). The TB program has been relatively well funded through the Global Fund, USAID, and other major donors; this support has built capacity in the health care system to diagnose and manage the TB prevention and control program. The relative improvement in the country's security situation also provided an opportunity to strengthen the TB program.

The national reporting system collected the results of routine contact screening for all 34 provinces from 2011 to 2018, following the implementation policy of the NTP. This study presents this programmatic experience, including only the public sector since the private sector was not engaged in contact screening and IPT administration. Before 2011, there was no contact screening policy or reporting. The national guidelines recommend both active and passive contact screening of the household members of bacteriologically confirmed index TB patients [10]. After a new index TB patient is identified and registered for treatment, TB experts counsel the index patient to bring his/her contacts to the health facility for screening, which is the passive screening approach, or a health worker or a community health worker (CHW) visits the house of the index TB patient to screen all household members, which is the active screening approach [10, 11].

The health worker, per the national guideline, asks each family member of the index TB patient if s/he has signs and symptoms of TB (productive cough for more than two weeks, weight loss, fever, and loss of appetite). Identified presumptive TB cases then give spot sputum, which is transported to an acid-fast bacilli (AFB) smear microscopy center on the same day. For the morning and spot sputum collection, the person with presumptive TB is asked to go to the health facility. At the health facility, people with presumptive TB are asked again about symptoms and signs, receive physical exams, and give sputum. If the sputum smear or GeneXpert test is negative, but there is a high degree of suspicion of TB or another disease, other investigations, such as chest x-rays, are undertaken. Per the national guidelines, two smears should be positive to support a diagnosis of TB. If only one smear out of three is positive, the doctor decides whether other investigations are warranted and starts treatment with a broad-spectrum antimicrobial. If abnormalities are present in the chest x-ray and the symptoms persists after antibiotic treatment, the patient is diagnosed as having pulmonary TB and anti-TB treatment starts. Extrapulmonary TB is diagnosed with a combination of histology, cytology, x-ray investigations, and clinical decisions [10].

The presumptive TB cases diagnosed with TB are enrolled for treatment, and children under the age of five in whom active TB has been ruled out are started on IPT for six months. IPT administration is demonstrated to the index TB patients, who administer IPT to children at home, and verified by CHWs when they provide directly observed treatment (DOT) to the index TB patient at home.

## Data collection and analysis

We collected the data from the NTP's annual and quarterly reports for 2011–2018. The national database since 2011 reports the number of all forms of TB diagnosed, number of bacteriologically confirmed TB cases, number of contacts screened, screening results (i.e., presumptive TB cases), number of children under five registered and put on IPT, and IPT completion rate. The database does not disaggregate the number of contacts screened by age for all years, so this information is not included in the analysis. The sex difference in performance was analyzed to see if there is a gender difference in services as well as disease burden, in order to inform future recommendations. The NTP, provincial health offices, and NGOs supporting the TB program carried out quarterly data quality assurance, and the quality of data has

improved over time. Individual data were entered into registers, and health facilities reported indicator-level aggregated data to provinces, which in turn aggregate the data and report it to the NTP manually. In Afghanistan there are two CHWs, one male and one female, for a population of 1,000–1,500, and most DOT is administered by CHWs. We aggregated the data by year and calculated the number needed to screen (NNS) to obtain a presumptive TB case and number needed to test (NNT) to diagnose a single case of TB. We calculated the 95% confidence intervals and *p* values using OpenEPI Software (Dean AG, Sullivan KM, Soe MM, OpenEpi: Open Source Epidemiologic Statistics for Public Health, Version 3.01, available from www.OpenEpi.com, updated 6 April 2013, accessed 23 Oct. 2019) and MedCalc software (Ostend, Belgium, Version 19, available from MedCalc.net, updated Aug. 2019).

### Definition of terms

We defined "bacteriologically confirmed index TB case" as a patient diagnosed with AFB sputum-smear microscopy as positive or any patient diagnosed positive with GeneXpert. A household contact per WHO's definition is "A person who shared the same enclosed living space for one or more nights or for frequent or extended periods during the day with the index case during the 3 months before commencement of the current treatment episode" [3]. A presumptive TB case is defined as having cough for two weeks or more, night sweating, and loss of weight, and, for children, failure to thrive. The NNS is defined as the number of contacts that have to be screened to detect a single case of presumptive TB, and the NNT is the number of presumptive TB cases that have to be evaluated to diagnose a single TB patient. IPT completion is defined as the completion of more than 95% of the prescribed dose of isoniazid, ascertained by TB focal persons by counting the pills in a blister pack, and DOT by CHWs when they observe the treatment of the index case.

### Ethical considerations

We obtained approval from the Afghanistan NTP to use the data collected and aggregated in the MOPH Management Information System database, and no additional ethical approval was required because we used de-identified data collected from routine programmatic reports of the NTP.

### Results

We analyzed a total of 298,332 TB cases between 2011 and 2018, and of those 142,797 (47.8%) were bacteriologically confirmed TB cases. Those bacteriologically confirmed TB cases were used as the index TB patients for the household contact screening. Based on data from the national census about average household size, the number of household contacts for each index TB patient was estimated to be six [12].

### The yield of TB screening

From 2011 to 2018, 586,292 household contacts were screened for TB, of whom 291,995 (49.8%) were females. Out of those screened, 117,643 (20.1%, 95% CI 19.96–20.07) were identified as presumptive TB cases. The presumptive TB cases identified totaled 55,255 (18.8%) in males and 62,388 (21.3%) in females ($p < 0.0001$). A total of 10,896 (1.8%, 95% CI 1.82–1.89) TB cases were diagnosed among all those screened, and out of those in whom TB (all forms) was diagnosed, 6,271 (57.5%, 95% CI 56.6–58.5) were bacteriologically confirmed TB patients (Table 1 and Fig 1).

**Table 1. Trends and yield of household contact investigation.**

| Year | All TB Cases Notified | Bacteriologically Confirmed TB Cases (%) | Estimated No. HH Contacts to Be Screened[a] | No. (%) HH Contacts Screened | No. (%) Presumptive TB Cases | No. (%) TB (All Forms) Diagnosed among HH Contacts | No. (%) Bacteriologically Confirmed TB among HH Contacts |
|---|---|---|---|---|---|---|---|
| 2011 | 28,167 | 15,103 (53.6) | 90,618 | 44,259 (49) | 16,145 (36.4) | 822 (1.8) | 606 (73.7) |
| 2012 | 29,578 | 14,464 (48.9) | 86,784 | 44,766 (52) | 7,939 (17.7) | 610 (1.3) | 350 (57.3) |
| 2013 | 31,622 | 15,670 (49.5) | 94,020 | 48,122 (51) | 8,274 (17.2) | 788 (1.6) | 390 (49.4) |
| 2014 | 32,712 | 16,182 (49.4) | 97,092 | 53,189 (55) | 9,403 (17.6) | 770 (1.4) | 469 (58.3) |
| 2015 | 37,001 | 17,975 (48.5) | 107,850 | 61,678 (57) | 13,367 (21.7) | 1,062 (1.7) | 580 (54.6) |
| 2016 | 43,046 | 19,948 (46.3) | 119,688 | 85,755 (72) | 15,569 (18.1) | 1,544 (1.8) | 894 (57.9) |
| 2017 | 47,406 | 21,316 (44.9) | 127,896 | 125,350 (98) | 22,499 (17.9) | 2,443 (1.9) | 1,511 (63.4) |
| 2018 | 48,800 | 22,139 (45.3) | 132,834 | 123,173 (93) | 24,447 (19.8) | 2,857 (2.3) | 1,471 (51.4) |
| **Total** | 298,332 | 142,797 (47.8) | 856,782 | 586,292 (81) | 117,643 (20.1) | 10,896 (1.8) | 6,271 (57.5) |

*Notes*: HH = household.

[a]The average number of household members in Afghanistan is reported to be six.

Out of all those screened, 4,963 (1.7%) of the males and 5,933 (2.03%) of the females were diagnosed with TB (all forms) ($p < 0.001$) (Table 2). In terms of yearly performance, the number of all forms of TB diagnosed progressively increased, from 28,167 in 2011 to 48,800 in 2018, which is a 73% increase (chi-square for trend = 269.2, $p < 0.01$). The number of bacteriologically confirmed cases also increased, from 15,103 in 2011 to 22,139 in 2018—an increase of 46.5%. Parallel to case detection, the number of index case household members screened also increased gradually, from 49% in 2011 to 93% in 2018 (chi-square for trend = 130.9, $p < 0.01$). The average yield of all forms of TB diagnosed was 1.8%, and it is only in 2018 that the rate was higher, at 2.3% (Table 1).

The proportion of presumptive TB cases identified was 18.8% (95% CI 18.63–18.92) for males and 21.4% (95% CI 21.2–21.5) for females ($p < 0.0001$). The NNS to identify a single presumptive TB cases was 53.8, and the NNT to diagnose a single case of all forms of TB was 10.8. The NNS for bacteriologically confirmed TB cases was 93.5 and the NNT was 18.7 (Table 3).

### Isoniazid preventive therapy

Contact screening was used as an entry point for identifying eligible children under the age of five for IPT. Out of the 586,292 household members of index TB patients screened for TB from 2011 through 2018, 117,593 children under five were eligible for IPT. Of those, 101,084 (85.9%) were initiated on IPT and 69,273 (68.5%) completed treatment. IPT coverage increased from 73% of the eligible children in 2011 to 94% in 2018. IPT completion also improved from 46% in 2011 to 74% in 2018 ($p < 0.0001$) (Table 4). In terms of sex, 59,873 children (50.9%) were females, and the number of male and female children who completed the full six months of IPT were 34,293 (68%) and 34,980 (69%), respectively ($p < 0.005$).

### Discussion

The Afghanistan NTP increased the screening of household contacts of bacteriologically confirmed index TB patients from 49% in 2011 to 93% in 2018 ($p < 0.0001$). The overall number of presumptive TB cases identified was 20.1%, which is higher than the numbers of presumptive TB cases reported in Ethiopia (6.1% and 11%) [7,13], and the 3% reported in Pakistan [8], 7.5% in Accra, Ghana [9], and 3% in another Afghanistan study [7]. The yield of all forms of

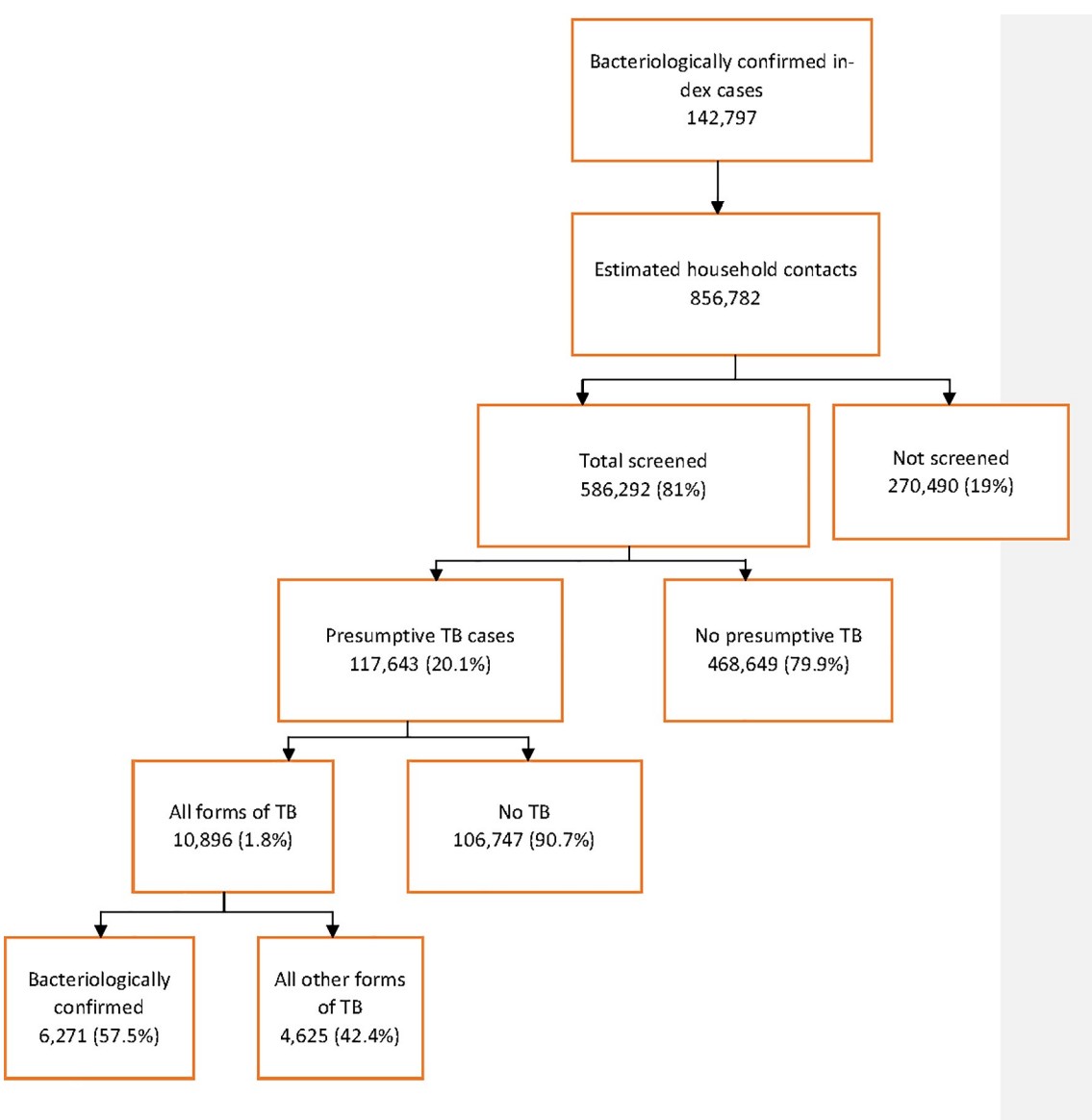

**Fig 1. Total screened, and yield of TB, Afghanistan (2011–2018).**

TB was 1.8%, which is close to 10 times higher than the estimated national incidence of TB, at 189 per 100,000, and is closer to the 1.6% reported in another study in Afghanistan [9] and the 1.7% reported in Viet Nam [5]. Furthermore, the yield of 1.8% found in this study was higher than the 0.65% in Ghana [14], 0.17% in Viet Nam [5], and 0.9% in Peru [6], but lower than the 3.6% that WHO reported [3], lower than the 3.9% reported in South Africa [15] and lower than the 2.5% reported in Ethiopia [7]. In a 21-year retrospective study in Birmingham, UK, the TB yield among smear-positive contacts screened was 7.0 [16]. The high number of presumptive TB cases identified in the routine program of Afghanistan is relatively low when considered as a proportion of all forms of TB diagnosed. It might be that health workers included people with signs and symptoms of TB but less than two weeks of history in screening; another reason for the relatively low yield could be the use of microscopy, a low-sensitivity diagnostic tool, in the majority of cases.

**Table 2. Yield of screening of household contacts of TB index cases by sex.**

| Year | No. HH Contacts Screened | | Ratio (F/M) | No. Presumptive TB Cases (%) | | No. TB (All Forms) Diagnosed among HH Contacts Screened (%) | | No. and Proportion of Bacteriologically Confirmed TB among all Forms of TB Diagnosed | |
|---|---|---|---|---|---|---|---|---|---|
| | **M** | **F** | | **M** | **F** | **M** | **F** | **M** | **F** |
| 2011 | 21,665 | 22,594 | 1.04 | 7,554 (34.9) | 8,591 (38.0) | 340 (1.6) | 482 (2.1) | 222 (65.3) | 384 (79.6) |
| 2012 | 22,339 | 22,427 | 1.00 | 3,713 (16.5) | 4,226 (18.8) | 253 (1.1) | 357 (1.6) | 139 (54.9) | 211 (59.1) |
| 2013 | 23,536 | 24,586 | 1.04 | 3,872 (16.5) | 4,402 (17.9) | 359 (1.5) | 429 (1.7) | 164 (45.6) | 226 (52.7) |
| 2014 | 27,422 | 25,767 | 0.94 | 4,372 (15.9) | 5,031 (19.5) | 359 (1.3) | 411 (1.6) | 192 (53.4) | 277 (67.3) |
| 2015 | 30,746 | 30,932 | 1.01 | 6,189 (20.1) | 7,178 (23.2) | 485 (1.6) | 577 (1.9) | 235 (48.4) | 345 (59.8) |
| 2016 | 43,142 | 42,613 | 0.99 | 7,312 (16.9) | 8,257 (19.4) | 669 (1.5) | 875 (2.0) | 353 (52.7) | 541 (61.8) |
| 2017 | 64,072 | 61,278 | 0.96 | 10,619 (16.6) | 11,880 (19.4) | 1,161 (1.8) | 1,282 (2.1) | 673 (57.9) | 838 (65.3) |
| 2018 | 61,375 | 61,798 | 1.01 | 11,624 (18.9) | 12,823 (20.7) | 1,337 (2.2) | 1,520 (2.5) | 672 (50.2) | 799 (52.5) |
| **Total** | 294,297 | 291,995 | 0.99 | 55,255 (18.8) | 62,388 (21.4) | 4,963 (1.7) | 5,933 (2.0) | 2,650 (53.4) | 3,621 (61.0) |

**Table 3. Yield of contact screening by sex.**

| Characteristics | Male | Female | Total | P Value |
|---|---|---|---|---|
| *Total contacts screened* | **294,297** | **291,995** | **586,292** | |
| *Presumptive TB cases* | 55,255 | 62,388 | 117,643 | |
| Proportion | 18.8 (18.63–18.92) | 21.4 (95% CI 21.2–21.5) | 20.0 (95% CI 19.96–20.17) | |
| NNS | 5.3 | 4.6 | 4.9 | |
| *All forms of TB* | **4,963** | **5,933** | **10,896** | |
| Proportion | 1.6 | 2.0 | 1.8 | < 0.0001 |
| NNS | 59.2 | 44.1 | 53.8 | |
| NNT | 11.1 | 10.5 | 10.8 | |
| *Bacteriologically confirmed total* | **2,650** | **3,621** | **6,271** | |
| Proportion | 53.4 | 61.0 | 57.5 | < 0.0001 |
| NNS | 111.0 | 80.6 | 93.5 | |
| NNT | 20.8 | 17.2 | 18.7 | |

**Table 4. IPT enrollment and completion rate for children under age five.**

| Year | No. HH Contacts under 5 Years Eligible for IPT | No. (%) HH Contacts under 5 Years Who Started IPT | No. (%) HH Contacts under 5 Years Who Completed IPT |
|---|---|---|---|
| 2011 | 8,534 | 6,199 (73%) | 2,823 (46%) |
| 2012 | 8,934 | 7,461 (84%) | 4,808 (64%) |
| 2013 | 10,620 | 7,690 (72%) | 4,924 (64%) |
| 2014 | 11,919 | 8,792 (74%) | 6,046 (69%) |
| 2015 | 11,799 | 10,164 (86%) | 7,737 (76%) |
| 2016 | 17,215 | 15,417 (90%) | 10,387 (67%) |
| 2017 | 24,656 | 22929 (93%) | 16,049 (70%) |
| 2018 | 23,916 | 22,432 (94%) | 16,499 (74%) |
| **Total** | 117,593 | 101,084 (85.9%) | 69,273 (68.5%) |

The other factor that should be studied is the lack of clear international guidelines about the definition of a contact. Contact screening is currently limited to the household members living under one roof, but in a traditional society like Afghanistan's extended family members have very close daily interactions. In a study in Pakistan, screening of household contacts and neighbors within 50 m of the radius of the index TB cases yielded rates of 22.3% and 19.1% of all forms of TB, respectively [8]. Given similarities between the community structure in Pakistan and Afghanistan, we recommend including neighbors in contact screening.

We found a significant difference between females and males ($p < 0.0001$) in the number of all forms of TB diagnosed, which is consistent with another Afghanistan study in which the diagnosed female-male ratio was 2.1 [9], while studies in Tanzania and England (London) did not show any difference between males and females [17–18].

The NNS to obtain a presumptive case of TB (all forms) was 53.8 and the NNT was 10.7. The NNS is lower than the NNS of 424 and 378 reported in the Afghanistan study [9]. In Ethiopia, the NNS and NNT were 40 and 2.4, respectively, and in Ghana, the NNS was 154 and the NNT was 8, respectively [7,14]. The NNT depends mainly on the incidence of the disease in a country, the quality of screening, and the screening tool used, so the differences among countries can be explained by these factors, but we do not know why the NNT in the other Afghanistan study was so high [9]. A different geography with a different disease burden or the capacity of the professionals involved in TB screening and diagnosis could be factors. The introduction of GeneXpert in recent years might also be a factor in the difference in results between our study and the other Afghanistan study.

The rate of bacteriologically confirmed cases in this analysis was 57.5%, which is close to the 61% reported by WHO [1]. The yield could have been higher if x-rays had been used as a screening tool and GeneXpert used as a diagnostic tool for all. As an example, in a study in Zambia, out of all contacts screened with chest x-ray, 53.6% had abnormal chest x-rays, and out of those, TB (all forms) was diagnosed in 32%; 19% of them were bacteriologically confirmed, but 8% had no symptoms [19]. Chest x-ray is highly sensitive, and most of the TB cases identified in national prevalence surveys and focal studies were found by chest x-ray rather than by symptomatic screening alone [17–20]. GeneXpert is also a more sensitive and specific tool for TB diagnosis [20,21]. In the future, the introduction of chest x-ray and GeneXpert for screening and diagnosis of TB, respectively, will help to increase case detection in Afghanistan.

Contact screening was used as an entry point for IPT initiation of children under age five who were contacts of bacteriologically confirmed TB patients. According to the 2018 WHO report, Afghanistan reported 100% initiation of IPT for the year 2017 [1], while we report 93% because of updates after the data were submitted to WHO. Afghanistan is one of the few countries with a high TB incidence rate to achieve this result [1]. The completion rate for six months of IPT is also very high, with an average of 68.5% from 2011 through 2018. To our knowledge, there is no nationwide report of completion of treatment for latent TB infection on this scale except from Mozambique [1], and the reports for various smaller groups ranged from 6% to 94% [22]. We attribute the high rate of completion to the counseling of parents on the importance of IPT, including health workers' encouragement of parents to complete children's IPT; good follow-up with patients who missed a day of treatment; and uninterrupted supply of isoniazid. The mandatory implementation of contact registration and screening of all index TB patients in Afghanistan increased case notification and served as an entry point to find IPT-eligible children. Another advantage is that the treatment of index TB patients is observed daily, so the health worker can ask them to bring all contacts for screening. CHWs visit index TB patients at home, providing another opportunity to screen all contacts and to supervise IPT.

## Limitations

Because the data for this study were collected from routine reports, and health facility registers do not include the number of household members, the denominator used to calculate the target for screening was the national average household size. Using that number might have inflated or underestimated the actual number of contacts screened. Furthermore, GeneXpert has been used since 2015 as a diagnostic method in a very small number of health facilities, but this study did not capture the results by diagnostic type for bacteriologically confirmed TB cases.

## Conclusions

This study found that contact screening of household members of TB index cases in Afghanistan is very high by global standards, and the yield of TB is also close to 10 times higher than the national estimated TB incidence from 2011 through 2018 [1]. We recommend following contacts for two to three years, since most contacts who develop TB disease do so within this time period. IPT initiation and completion rates are very high, and along with the extension of contact screening to neighbors, IPT targets should also be revised. Shifting from IPT to rifapentine- or rifampicin-based shorter TB preventive treatment for latent TB infection [23] will further improve treatment completion, and the country should revise its policy to extend preventive treatment to all eligible contacts of all ages.

## Supporting information

**S1 File.**
(XLS)

## Acknowledgments

We thank the teams from the NTP and Challenge TB Afghanistan for their support in the development of this paper. Erik J. Schouten assisted in reviewing the draft article, and Barbara K. Timmons edited and formatted the article. We also appreciate the support of the home office team of Management Sciences for Health.

## Author Contributions

**Conceptualization:** Said Mirza Sayedi, Ghulam Qader, Pedro G. Suarez.

**Formal analysis:** Ghulam Qader, Muluken Melese.

**Investigation:** Said Mirza Sayedi, Ghulam Qader, Muluken Melese.

**Methodology:** Said Mirza Sayedi, Ghulam Qader, Muluken Melese.

**Project administration:** Said Mirza Sayedi.

**Supervision:** Mohammad K. Rashidi.

**Visualization:** Muluken Melese.

**Writing – original draft:** Said Mirza Sayedi, Muluken Melese, Pedro G. Suarez.

**Writing – review & editing:** Said Mirza Sayedi, Mohammad Khaled Seddiq, Mohammad K. Rashidi, Ghulam Qader, Naser Ikram, Muluken Melese, Pedro G. Suarez.

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
