## [Decision Letter · Decision Letter 0]

17 Dec 2019

PONE-D-19-30098

Active household contact screening for tuberculosis and provision of isoniazid preventive therapy to children in Afghanistan

PLOS ONE

Dear Dr Sayedi,

Thank you for submitting your manuscript to PLOS ONE. After careful consideration, we feel that it has merit but does not fully meet PLOS ONE’s publication criteria as it currently stands. Therefore, we invite you to submit a revised version of the manuscript that addresses the points raised during the review process.

We would appreciate receiving your revised manuscript by Jan 31 2020 11:59PM. To enhance the reproducibility of your results, we recommend that if applicable you deposit your laboratory protocols in protocols.io, where a protocol can be assigned its own identifier (DOI) such that it can be cited independently in the future. For instructions see: http://journals.plos.org/plosone/s/submission-guidelines#loc-laboratory-protocols

We look forward to receiving your revised manuscript.

Kind regards,

Philip C Hill, MD

Academic Editor

PLOS ONE

Journal Requirements:

Additional Editor Comments (if provided):

Please address all the comments by the reviewers. Please note, the paper will need to be improved both scientifically and in grammatical aspects.

Reviewers' comments:

Reviewer's Responses to Questions

**Comments to the Author**

1. Is the manuscript technically sound, and do the data support the conclusions?

Reviewer #1: No

Reviewer #2: Yes

2. Has the statistical analysis been performed appropriately and rigorously? 

Reviewer #1: No

Reviewer #2: Yes

3. Have the authors made all data underlying the findings in their manuscript fully available?

Reviewer #1: No

Reviewer #2: Yes

4. Is the manuscript presented in an intelligible fashion and written in standard English?

Reviewer #1: No

Reviewer #2: Yes

5. Review Comments to the Author

Reviewer #1: This is an interesting study of secondary data on contact investigation, a high priority according to the UN high-level meeting held in NYC in 2018. Two previous studies have reported the prevalence of active TB among contacts in Afghanistan. The current manuscript adds a lot to these reports, by discussing LTBI treatment, uptake and completion, and by describing the substantial increase in case finding after the implementation of a new policy in the country. These findings are interesting. The manuscript, however, needs some improvements before it can be suitable for publication.

Abstract:

This abstract should be rewritten. The objectives should clearly state that a new policy was recommended by the NTP in 2011 for contact investigation and this observational study reports the findings since these recommendations.

The Methods section currently describes mainly active TB indicators (NNS and NNT) but the conclusions focus on IPT in children. The authors need to choose what their main message is and write a coherent abstract.

If the number of children starting and completing LTBI treatment is their main focus, they need to define what the national policies for contact investigation and treatment for LTBI were in 2011-2018 in the Methods section, so that the reader can understand why so many children under 5 were prescribed IPT.

In the conclusion, what do the authors call a “high” performance? Do they mean good? The findings should be interpreted in the same was they are reported in the Results section. Incidence rate is not shown in results, so the reader can not follow the conclusion. Please adjust.

The conclusion should also describe briefly how the authors explain the high rates of IPT prescription and completion when compared to other countries.

The sentence ending the abstract is misleading: Afghanistan is not a HBC, and the sentence refers to “other” HBC. Afghanistan is a high TB incidence RATE country, which is different from HBC. Please adjust.

The short title should be adjusted. The study is not about children only.

Main text:

Introduction:

This report is of interest for many other countries. Accordingly, the introduction should highlight why the investigation of contacts is important (anywhere), and just a short paragraph to explain why it is important in Afghanistan. The detailed description of the TB situation in Afghanistan should be reported in a “setting” subheading of the Methods section.

L. 32-33: Incidence and incidence rates are different concepts, please rephrase.

Methods

This is an observational study of a new policy recommended by the NTP. The new policy and the routine investigation of contacts should be explained but not be named “intervention” because this is misleading. All that is currently reported as the “intervention” should be reported as setting. This section should also make clear what were the previous recommendations and what is new in the current recommendation.

Are the reported definitions the ones adopted by the NTP? Are health care workers and community agents instructed to identify contacts as per this definition?

Data analysis: time trend would be a better analysis (quarterly, since these data is available by quarter) than proportions. The increase (or any change) in the population (denominator) should be taken into consideration in analyses/comments of incidence rates.

Ethics: how was data stored? What procedures were made to protect the data?

Results:

“We included” is not a good term for an analysis of secondary data.

The p-values are unnecessary, since the CI are presented (and any difference with such a huge sample would be “statistically significant”.

Discussion

The results should not be repeated in the discussion.

Instead of simply comparing the yield of active TB among screened contacts, it would be interesting to discuss what might explain these differences.

What does this manuscript add to the two previous reports on contact investigation in the country? This should be highlighted in the discussion. How do the current findings compare to findings before the new policy? In other words, what is the impact of the new policy? Were there any other changes in the country or health system in this period that could explain the increase in the proportion of TB cases diagnosed?

Comments on cost-effectiveness of the NNS and NNT would be interesting, even if speculative since this kind of analysis was not the scope of the study.

Above all, the conclusions cannot be withdrawn from the findings of the study.

Minor comments:

1. The text needs some language review.

2. References need to be updated.

Reviewer #2: Summary of Paper:

The manuscript summarizes 8 years of national TB program data in Afghanistan. During this time period 142, 797 active TB cases were reported and 586,292 household contacts of index cases were screened for TB. The number of co-prevalent cases among HHC was found to be high. TPT initiation and completion rates for children under five were high. In many cases data is reported separately for men and women. The number needed to screen (NNS) and number needed to treat (NNT) are also reported.

General comments:

Overall this was an informative and well written paper. Data is presented for many useful TB indicators. The performance of the program is remarkable. Some additional explanation as to how the program has been able to perform so well would be useful so that others can learn from the experience in Afghanistan.

Major comments:

1) On page 4 the authors state that they use “routine contact screen performance” data from the Afghanistan NTP from 2011-2018. It would be useful to know more about how this data is collected and how good the data is felt to be? (has the same registry been in place since 2011 to collect this data? Are they electronic registries? Do they combine data from multiple registries? How reliable are the data collected in screening registries? How complete are they?)

2) Related to the point above, within these registries how is treatment completion measured?

3) How large is the private sector in Afghanistan? If the private sector is involved in TB care, how is this captured in the data reported? Is the private sector involved in routine program reporting? The role of the private sector and the generalizability of the findings reported in the manuscript data should be included in the discussion

4) The authors show that there have been major improvement in the TB outcomes reported during this period. It would be interesting to know what mechanisms may have contributed to the major improvements seen in the country in the last few year. For example has program funding remained stable over these years? How/why has it changed? Have strengthening projects been in place? Readers will be interested to know what investments are required in order to bring a program to this level (both in terms of both $ and program strengthening through Challenge TB etc…)

5) Throughout the manuscript, the authors stratify most findings by sex, and use statistical testing to look at sex differences. It would be helpful to provide a justification in background to support these analyses. Was there an initial hypothesis that there would be gender differences which led to these analysis? I’m not sure what the statistical testing adds without a clear discussion of the rationale and implications. In the discussion there is currently just a comparison to what others have found.

6) In the paper the authors use census data to estimate 6 household members. Do the authors know how many HHC were usually identified when contact investigators were performed? Is there any data available to validate the census assumption?

7) In the discussion the authors state that they attribute the high rate of completion to various things (counseling of parents on the importance of IPT etc..). It would be nice to have more detail about what is involved programmatically in achieving this level of passive case detection, screening, testing and treatment initiation, so that others can learn from the successes of this program. For example, what is involved in good follow up? What lengths do HCW go to in order to help people initiate and complete IPT. If the authors can elaborate on any of these points it would strengthen the paper.

8) The authors final statement that IPT “should probably be extended to all eligible contacts of all ages” seems like a bit of an afterthought. For a program that is seemingly performing so well, one could argue this would be an obvious next step. Can the authors comment on if there are plans for this, or if there are obstacles? It seems that this notion deserves more attention, especially in light on the UNHLM targets to expand TPT to 20 million HHC of all ages by 2022.

.

Minor points:

9) line 110- it should be made clear that by “contact” they are referring to household contacts (based on the definition provided). Throughout would suggest referring to household contact for clarity.

10) line 114 there is a typo?- should be NNS (not NNT)

11) Line 235- typo? It states “interrupted” supply of INH- presumably it should say “uninterrupted”.

12) Page 6 the first line on page 7 (line 132) is redundant as it has already been stated in the first line of the results

13) Page 13, line 251- we recommend following contacts for 2-3 years- to monitor for TB? Pls clarify for what reasons

6. PLOS authors have the option to publish the peer review history of their article (what does this mean?). If published, this will include your full peer review and any attached files.

Reviewer #1: No

Reviewer #2: No

---

## [Author Response · Author response to Decision Letter 0]

4 Aug 2020

I have attached a letter with response to comments for reviewers as revised submission document

---

## [Editor Report · Decision Letter 1]

25 Aug 2020

PONE-D-19-30098R1

Active household contact screening for tuberculosis and provision of isoniazid preventive therapy to under-five children in Afghanistan

PLOS ONE

Dear Dr. Sayedi,

Thank you for submitting your manuscript to PLOS ONE. After careful consideration, we feel that it has merit but does not fully meet PLOS ONE’s publication criteria as it currently stands. Therefore, we invite you to submit a revised version of the manuscript that addresses the points raised during the review process.

We look forward to receiving your revised manuscript.

Kind regards,

Philip C Hill, MD

Academic Editor

PLOS ONE

Additional Editor Comments (if provided):

I am afraid the quality of the English, especially in the newly revised sections is not good enough for publication

I am happy to give you an opportunity to fix this issue.

---

## [Author Response · Author response to Decision Letter 1]

15 Sep 2020

all the data are in the article. 

I have attached two documents for reviewers comments

The English editing done

An excel file with data attached

---

## [Editor Report · Decision Letter 2]

18 Sep 2020

Active household contact screening for tuberculosis and provision of isoniazid preventive therapy to under-five children in Afghanistan

PONE-D-19-30098R2

Dear Dr. Sayedi,

We’re pleased to inform you that your manuscript has been judged scientifically suitable for publication and will be formally accepted for publication once it meets all outstanding technical requirements.

Kind regards,

Philip C Hill, MD

Academic Editor

PLOS ONE
---

## [Editor Report · Acceptance letter]

30 Sep 2020

PONE-D-19-30098R2 

Active household contact screening for tuberculosis and provision of isoniazid preventive therapy to under-five children in Afghanistan

Dear Dr. Sayedi:

I'm pleased to inform you that your manuscript has been deemed suitable for publication in PLOS ONE. Congratulations! Your manuscript is now with our production department. 

Kind regards, 

on behalf of

Prof. Philip C Hill 

Academic Editor

PLOS ONE